### Seasonal, Spring-Neap and Tidal Variation of Hydrodynamics and Water Constituents in the Mouth of the Elbe Estuary, Germany

Jens Kappenberg<sup>1</sup>, Michael Berendt<sup>2</sup>, Nino Ohle<sup>2</sup>, Rolf Riethmüller<sup>1</sup>, Dagmar Schuster<sup>2</sup>, and Thomas Strotmann<sup>2</sup>

<sup>1</sup>Helmholtz-Zentrum Geesthacht, Max-Planck-Strasse 1, 21502 Geesthacht, Germany
 <sup>2</sup>Hamburg Port Authority, Neuer Wandrahm 4, 20457 Hamburg, Germany
 *Correspondence to*: Jens Kappenberg (jens.kappenberg@hzg.de)

#### Abstract

Results of long-term measurements of CTD, current velocity, turbidity, and dissolved oxygen from three stations in the

- mouth of the Elbe Estuary in northern Germany are presented for the period of 2012 and 2013. The focus is on a station named HPA-Elbe 1, which is part of the COSYNA coastal observing system for northern and arctic seas, but data from two neighbouring stations are also presented for comparison and to bridge gaps in the time-series of HPA-Elbe 1. The variations of the variables exhibit distinct tidal patterns related to the longitudinal estuarine gradients of the water constituents and local sediment dynamics. On longer time-scales, spring-neap variability is observed, most prominently in the hydrodynamics. On
- the seasonal scale, the water temperature influences the degradation processes, which deplete the dissolved oxygen on the one hand and increase the oxygen contents by biological respiration on the other hand. Freshwater events from the catchment play an important role for sediment dynamics and mixing of riverine and coastal waters in the brackish water reach of the estuary. The periods of the deployment of the stations comprise the severest river flood observed so far in the Elbe in June 2013. The effects of the flood at the stations and along the estuary consist of a rapid flushing of the mixing zone and the
- turbidity maximum to the outer estuary and the transition to freshwater conditions from Hamburg to the mouth at Cuxhaven. The impact of the river flood at the stations is more pronounced in changes in salinity than in turbidity. The restoration of estuarine salinity and turbidity values comparable to those before the river flood took several months.

#### Keywords

Long-term measurements 25 Estuarine processes Monitoring stations Tidal, spring-neap, seasonal variabilities

# Ocean Science

#### 1 Introduction: the Elbe Estuary

- Estuaries are extremely variable systems dependent on strongly varying external forcings. The freshwater input from the catchment, for example, for many estuaries shows a seasonal variation over one to two orders of magnitude. To assign a characteristic state or a classification to a particular estuary is therefore no trivial task and the estuarine classification schemes the classes have somewhat fuzzy borders (see Dyer, 1977). The importance of long-term measurements of physical variables in estuaries resides in the possibility to analyze on different time-scales the ongoing precesses and the complex
- interactions of turbulence, stratification, advection, and mixing. In this paper long-term time-series from some stations in the in the mouth of the Elbe are investigated. These are point measurements with no vertical and horizontal resolution. So they will not shed new light on the current front edges of estuarine physical research like exchange flows (estuarine circulation), vertical and transverse hydrodynamics (seccondary circulation) and estuarine fronts as reviewed for the Elbe by Burchard and Baumert (1998) and on a global scale by McCready and Geyer (2010), Uncles (2002), and Uncles et al. (2002). The attempt is made to integrate the station measurements into the general estuarine context by comparing them to data from
- In all ongitudinal near-surface profiles of the estuary undertaken four to five times a year at a fixed tidal phase. The Elbe Estuary is located in Northern Germany. It extends from mouth at Cuxhaven to the tidal border at Geesthacht covering a total length of about 120 km (Fig. 1). It is the largest estuary on the German coast of the North Sea and an important waterway connecting the Port of Hamburg, which is third largest in Europe with the sea. According to Dyer (1973)
- it can be categorized as a partially mixed coastal plain estuary, The large tidal range (4m at spring tide) places the Elbe at the border between meso- and macrotidal estuaries (Boehlich and Strotmann, 2008). Tides are semidiurnal with a marked diurnal asymmetry. Due to the combined effects of bathymetry (navigational channel to Hamburg) and shape changes (convergence) of the cross-sections the mean tidal range decreases from 3m at the inlet to 2.7m 50 km up-estuary, rises again towards the port of Hamburg, where it attains a maximum of 3.7m, and falls to 2m at the weir. The Elbe drains 132,000 km<sup>2</sup> of Central
- Europe. The long-term average freshwater runoff at Neu-Darchau (53 km upstream of tidal limit) is about 715 m<sup>3</sup> s<sup>-1</sup>, the riverine influx of suspended particulate matter (SPM) is about 750,000 tons per year (DGJ, 2013). The morphological diversity of the Elbe Estuary is mainly governed by tidal action and is naturally characterised by substantial sediment transport linked to the continuous remodelling within the estuary. Extensive sediment relocation takes
- place due to erosion and re-sedimentation enhanced by heavy storm surges. Characteristic features of natural estuaries include multiple and migrating channel systems, changing river width, scours and aggradations, inter-tidal areas, as well as the formation of sand banks and islands.

A prominent feature of the Elbe Estuary is the estuarine turbidity maximum (ETM), usually located some 40 km from the inlet and extending over more than 30 km. SPM concentrations (SPM) in this area exceed the riverine values  $(SPM_{mean} = 35 \text{ g m}^{-3})$  by a factor of 10 to 30. The location of the ETM can vary up to 25 km depending on river runoff and

tidal conditions. Bergemann (1995) argues that the ETM has moved upstream between 5 and 20 km from 1953 to 1994. The upstream end of the mixing zone (MZ), where the salt water from the North Sea mixes with the riverine freshwater is located

near Glückstadt some 50 km from the mouth. Similar to the ETM it has moved further upstream during the last few decades. This is due to the widening cross-section at the mouth and engineering works like the on-going deepening of the navigational channel and flood-protection within the estuary.

Despite its importance for shipping and the environment there are few published results of investigations into the physical estuarine processes in the Elbe. On the other hand there is a growing demand for the understanding of these processes to facilitate sediment management in the context of European directives like the WFD and the FFH and to establish understanding with local stakeholders for interventions into the estuarine system.

#### 2 Measurement Stations and Instrumentation

#### 70 2.1 Location

The data discussed here are from four long-term time-series taken in the Elbe Estuary (Fig. 1) during the years 2012 and 2013. Three of them were collected at fixed stations mounted close to the Elbe mouth. A pole system, named HPA-Elbe 1, was operated jointly by Hamburg Port Authority (HPA) and Helmholtz-Zentrum Geesthacht (HZG) as part of the COSYNA research project (Baschek et al., 2016). HPA-Elbe 1 is placed at the southern edge of a wadden area, the Neufelder Watt,

- which has been extensively investigated by HPA in previous years (Albers et al. 2009; Albers, 2012), because it had been hypothesized that significant sediment input occurs from this area into the navigational channel. In the estuarine context, the position HPA-Elbe 1 is in the seaward part of the mixing zone and near the seaward end of the ETM. This holds for mean conditions since both the position of the MZ and the ETM are subject to considerable variations mainly due to the freshwater discharge of the Elbe River. The two other stations, LZ2 and MPM Otterndorf were operated by the water authorities to
- monitor the effects of the last adaptation of the navigational channel in 1999 (WSV, 2016). HPA-Elbe 1 and LZ2 were located at the northern and MPM Otterndorf at the southern boundary of the navigation channel. In addition to these point measurements data from surface samples taken by helicopter on nine transects of the estuary are used to discuss the station measurements in the overall estuarine context (FGG Elbe, 2016). The samples are taken at maximum ebb current, assuming maximum mixing of the water column, and the helicopter allows for keeping the phase of the tide and sampling in the
- navigational channel. The length of the transect is 172 km from the German Bight to the weir at Geesthacht and samples are taken every 5 km.

#### Figure 1: The Elbe Estuary and location of the measurement stations in the mouth.

#### 2.2 Instrumentation and measured variables

The pole HPA-Elbe 1 (Fig. 2) was deployed in 2012 and 2013 from March to end of October (2013) / mid of November (2012) to prevent ice damage in the winter months. It consisted of a 15 m long steel tube of 40 cm diameter, 5 m of which are jetted into the seabed. A platform accessible via a ladder is mounted on top of the tube, resulting in an overall length of 18 m (Fig. 2). The platform carried meteorological sensors and radiometer, solar panels for energy supply, an automated yet

remotely controllable water sampler, and logger boxes for temporary data storage and wireless communication. A manual
winch was used to retrieve the underwater instrument unit for maintenance. The underwater unit was mounted so that the
lower end of its sensor package was positioned 1.5 m above the sea floor. It was equipped with sensors for conductivity
(AMD Elektronik), water temperature (PT100, AMD Elektronik), pressure (PA–7, Sea&Sun), current velocities (Nortek
Vector Acoustic Doppler Velocimeter), as well as optical sensors recording transmittance at 660 nm wave length (Sea&Sun),
turbidity at 880 nm (Seapoint), chlorophyll-a fluorescence (MicroFlu–chl, TRIOS) and oxygen saturation (Aanderaa Oxygen
Optode 4175). A tide and wave gauge floater was mounted laterally to the tube of the pole.

- The data from all instruments were sampled at 2 Hz and then averaged over 10 min due to limited bandwidth of the wireless communication. For periods with wave heights above selectable thresholds, the data was stored at high sampling frequency. In order to reduce sensor fouling, e.g. by seaweed, mussels, barnacles and other organic material, the underwater unit was cleaned at least twice a month. Possible sensor drift and cleansing effects were monitored by direct comparison with a wellcalibrated reference system before, during, and after maintenance.
  - Since the parameterization of the curve of the Elbe River is by kilometres (Elbe km, 0 km = German-Czech border) in downstream direction, the convention is adopted that ebb currents have a positive and flood currents a negative sign.

Figure 2: Pole HPA-Elbe 1 in the mouth of the Elbe Estuary with instrumentation and sensors.

#### 110 3. Periods of operation, hydrological and meteorological conditions

Figure 3 gives an overview on the periods during which data from the different stations are available in 2012 and 2013. In the top panel, the freshwater runoff at Neu-Darchau is displayed to indicate the hydrological situation during the measurements. 2012 was a year with rather low runoff, the annual mean amounting to  $635 \text{ m}^3 \text{ s}^{-1}$ . The hydrological situation in 2013 is characterized by two runoff events in the first two months and a strong river flood in June with a peak value of

115 4066 m<sup>3</sup> s<sup>-1</sup>, the highest ever recorded in the Elbe, raising the annual mean to 1014 m<sup>3</sup> s<sup>-1</sup>. In September 2012 a week of data is missing due to sensor failure. Data from LZ2 are also available during wintertime with occasional gaps due to failure of equipment. The station MPM Otterndorf was established in February 2013. In 2013 the river flood is monitored by all stations.

# 120 Figure 3: Freshwater discharge and periods of operation of the different stations. Vertical lines indicate dates of transects by helicopter.

Although the instrumentation of HPA-Elbe 1 comprises a meteorological station where wind speed and direction are recorded, wind data from the island of Scharhörn (Fig. 1) some 35 km to the west of HPA-Elbe 1 are used as boundary

conditions for the measurements. This is because of the lack of data on HPA-Elbe 1 during the winter season. In figure 4 the wind statistics for Scharhörn during 2012 and 2013 are displayed in 10 degree wind direction intervals. The shaded grey area indicates the relative frequency of winds coming from a 10 degree direction interval for all wind speeds. Restricting this

analysis to wind speeds above 18 m s<sup>-1</sup> (high wind speeds) results in the coloured sectors. In 2012 and 2013 the wind was blowing mainly from westerly directions. The most frequent wind direction is a south-westerly coming from 210 degrees.
High wind speeds always come from westerly directions, while weak and moderate true easterly winds coming from the 90 degree sector are also quite abundant especially during summer. The strongest winds come from two sectors between 210 and 250 degrees and between 290 and 310 degrees.

Figure 4: Distribution of wind directions and wind speeds on the island of Scharhörn 35 km to the west of station HPA-Elbe 1 in 2012 and 2013

#### 4 Results

#### 4.1 Measurements at the HPA-Elbe 1 station

In figure 5 the time-series of the variables measured at HPA-Elbe 1 are displayed for a nine day episode during the river flood in June 2013. The top panel shows the current velocity in magnitude and direction. There is a remarkable tidal asymmetry and a clear ebb dominance as well in the magnitude of the current velocity as in the duration of the ebb-tide. At the location of the station in the outer estuary, one would expect a more symmetrical tide and the asymmetry is probably due to influence of the branch (Medem Rinne) to the Northwest of HPA-Elbe 1. The current direction indicates flow reversal by 180 degrees in the overall direction of the main channel and the edge of the wadden area to the North of the station. Tidal range during this episode is about 3 m, HW occurs about 1 hour before high water slack, and LW about 1 hour before the low

- water slack. Further, the water elevation time-series shows diurnal inequality. Water temperature varied between 16 and 19 degrees Celsius with maximum values often coinciding with peak ebb current velocities when warmer water from upestuary is passing the station. The effect of the river flood is most clearly pronounced from June 10 on in the measured salinities. Peak values at high water slack of 8 psu before the river flood are reduced to below 2 psu as freshwater is discharged into the outer estuary and the MZ pushed seaward. The minimum salinity at the end of the ebb-tide drops from
- 1.5 psu pre-event values to almost zero after the river flood. Before the flood the highest values of dissolved oxygen saturation are attained at high water slack when oxygen rich coastal water arrives at the station. During the river flood, oxygen saturation has a minimum before or around high water slack indicating the existence of oxygen depleted water masses down-estuary of HPA-Elbe 1. The effects of the event can also be detected in the turbidity. While at the beginning of the episode the maximum turbidity during flood-tide exceeds the ebb-tide maximum by a factor of 1.5, after the event
- generally lower turbidity values are observed and ebb and flood maxima are more balanced. Accidentally the maximum of the river flood on June 11 coincides with a change in wind direction from north-westerly to southerly on June 11-12.

Figure 5: Time-series of current velocity (grey shaded curve) and direction (solid black curve), water elevation (grey shaded curve), water temperature (solid black curve), salinity (grey shaded curve), dissolved oxygen (solid black curve), turbidity (grey shaded curve), and wind vector at HPA-Elbe 1 during the river flood in June 2013. Vertical lines indicate times of tidal current reversal at high water slack (dashed line) and at low water slack (solid line).

In general, the occurrence of quasi-periodic intratidal variations of water constituents at a fixed estuarine station can be explained by the advection of a stationary longitudinal gradient of the constituent (salinity, temperature, and oxygen). In the case of turbidity, assumed as an approximately linear surrogate for SPM, local processes like settling and resuspension of material related to the periodic variation of current velocity are superimposed.

To evaluate the measurements at HPA-Elbe 1 on longer time scales tidal averages and extremes (tidal minimum and maximum values) have been calculated for full operation period. For this purpose a tide is defined as the time between successive high water slacks on the basis of the current velocity measurements. In figure 6 the time-series of tidal averages

- and extremes are depicted for wind speed, salinity and turbidity. The vertical numbered lines mark instants of 9 longitudinal helicopter transects (five in 2012 and four in 2013) of the estuary. Salinity and SPM data from the transects are shown in the lower panel. During March to June 2012 after the runoff events of the preceding months a general increase of salinity and a decrease of turbidity (SPM) is observed. From July onward to the end of the measurements in November 2012 the levels of tidal averaged salinity and turbidity stabilize at some 12 psu and 200 NTU with some variation connected to wind influence
- under uniform low runoff conditions. The influence of the wind can be exemplified at the time of transect 5 in the beginning of November 2012. With increasing westerly wind speeds saline water was pushed into the estuary and turbidity was reduced by an up-estuary transgression of the ETM.

While for salinity the tidal average is in the middle between the tidal extremes, for the turbidity data the tidal average (black curve) is always below the mean of tidal maximum and minimum. Temporal maxima of turbidity occur generally for short

time during flood-tide, but lower turbidity values during the longer ebb-tide contribute to a greater extent to the tidal average.

In the 2012 transects at the landward end of the salinity gradient a well developed ETM can be recognized, located between Elbe km 650 and 700. Considering that the transects are performed maximum ebb current the position of HPA-Elbe 1 is always on the seaward flank of the ETM in 2012.

Figure 6: Overview of the boundary conditions (runoff, wind speed) and measurements of salinity and turbidity at station HPA-Elbe 1. In the panels for wind speed, salinity, and turbidity tidal averages (solid black lines) and tidal extremes (grey shaded areas) are displayed. In the lowest panel near surface salinity (solid black line) and suspended particulate matter concentration (SPM, grey shaded curve) along the estuary from the 9 helicopter transects are shown. The vertical line marked HPA indicates the position of the station. The dates of transects are also indicated as vertical lines in the time-series plots in the upper panels.

The time-series of 2013 are dominated by the flood event in June. The first transect in May shows only a weak ETM, probably due to the high level of runoff in spring and winter. During the first weeks of June the rising river flood reduces the salinity and somewhat later turbidity at the station by pushing the saline riverine interface and the ETM to the German Bight. Transect number 7 on the peak of the event shows the ETM seaward of HPA-Elbe 1 and freshwater conditions up to

Cuxhaven. There is a small rise in turbidity at the end of May, which may be connected to riverine SPM transported downestuary on the rising edge of the river flood, a phenomenon that was observed at the head of the estuary (weir) during the 2013 and previous river floods. The duration of the flood effects at this site is different for salinity and turbidity. Already at

the beginning of July the turbidity regains the pre-river-flood level, while it took two months to re-establish the salinity 200 levels. The helicopter transects show only at the beginning of November the reestablishment of a fully developed ETM at its previous site. Figure 7 shows that salinity and freshwater discharge do not follow a simple exponential relationship: the same salinity value at HPA-Elbe 1 can occur during higher or lower river runoffs depending on whether the freshwater discharge during the period previous to the instant was higher or lower than the current value. This hysteresis of salinity as a function of runoff is displayed in the lower panel of figure 7.

Figure 7: Upper panel: Time-series of tide averaged salinity at HPA-Elbe 1 (solid black curve) and freshwater runoff at Neu-Darchau (shaded grey curve) from May to July 2013 containing the period of the 2013 river flood. Lower panel: Trajectory of the tide averaged salinity in runoff/salinity space. Numbers refer to the instants indicated as vertical black bars in the runoff timeseries. Note from figure 6 that at points 1 and 2 saline coastal water is pushed into the estuary by strong (westerly) winds.

Figure 8 gives an overview of the measurements of water temperature and dissolved oxygen saturation at HPA-Elbe 1 in the same manner as figure 6 does for salinity and turbidity. The helicopter transect data in the lowest panel of figure 8 show a typical oxygen sag curve down-estuary of Hamburg (Elbe km 620) during the warmer seasons with minimum oxygen saturations of below 50 percent. In general oxygen saturations along the transect were higher in 2013 than in 2012. The high

- (over-)saturations in the upper estuary were caused by biological oxygen production by algae and phyto-plankton. In autumn and winter the dissolved oxygen levels are higher and more homogeneous along the estuary. In the warmer season the river water is warmer than the coastal water, while the November transects show that the temperature gradient along the estuary is reversed in autumn and winter. The position of HPA-Elbe 1 is far down-estuary of the oxygen minimum and little of the dynamics of oxygen depletion can be observed at this site. By tidal excursion different parts of the longitudinal oxygen
- gradient are observed at the station, resulting in an oxygen time-series with tidal periodicity (Fig. 5) and the band between tidal extremes in figure 8. The tidal variation is largest when the station is located in the steepest part of the longitudinal oxygen gradient during May 2012 and 2013 and August 2013. Although a higher oxygen level in the estuary is indicated by the helicopter transects in 2013 than in 2012, the station data show somewhat reduced oxygen saturations in 2013 at HPA-Elbe 1.

Figure 8: Overview of the boundary conditions (runoff, wind speed) and measurements of water temperature and dissolved oxygen saturation (DO) at station HPA-Elbe 1. In the panels for wind speed, temperature, and dissolved oxygen saturation tidal averages (solid black lines) and tidal extremes (grey shaded areas) are displayed. In the lowest panel near water temperature (solid black line) and dissolved oxygen saturation (DO, grey shaded curve) along the estuary from the 9 helicopter transects are shown. The vertical line marked HPA indicates the position of the station. The dates of transects are also indicated as vertical lines in the time-series plots in the upper panels.

#### 4.2 Measurements at the LZ2 Station

LZ2 is located some 4 km up-estuary of HPA-Elbe 1, also at the northern boundary of the navigational channel. In figure 9 the corresponding data to HPA-Elbe 1 in figure 6 are displayed for LZ2. The time-series of the measured variables are more

complete than at HPA-Elbe 1 and include the winter half-year. At LZ2, situated up-estuary of HPA-Elbe 1 and closer to the ETM, salinity levels are some 2 psu lower and turbidity is generally higher than at HPA-Elbe 1.

In 2012 the effects of the runoff event at the end of January result in a decrease of salinity and turbidity caused by the downestuary movement of the salinity gradient and the ETM. As in the case of HPA-Elbe 1 from March to June 2012 an increase of salinity was accompanied by a decrease of turbidity, probably due to decreasing runoff. The next runoff event was at the

240 beginning of January 2013 with similar impacts as in 2012. Runoff remained relatively high during the following months, so salinity stayed lower than in 2012, while turbidity persisted at a high level with little variation. The impacts of the river flood in June 2013 were similar to HPA-Elbe 1, but the quick recovery of turbidity at HPA-Elbe 1 was not observed at LZ2. Salinity and turbidity levels regained their pre-river-flood levels only by August. On December 6 the effects of a severe storm surge can be seen in the wind speed and extreme salinity data.

Figure 9: Overview of the boundary conditions (runoff, wind speed) and measurements of salinity and turbidity at station LZ2. In the panels for wind speed, salinity, and turbidity tidal averages (solid black lines) and tidal extremes (grey shaded areas) are displayed. In the lowest panel near surface salinity (solid black line) and suspended particulate matter concentration (SPM, grey shaded curve) along the estuary from the 9 helicopter transects are shown. The vertical line marked LZ2 indicates the position of the station. The dates of transects are also indicated as vertical lines in the time-series plots in the upper panels.

#### 4.3 Measurements at the MPM Otterndorf station

MPM Otterndorf is the most down-estuary of the stations, located some 4 km from HPA-Elbe 1 on the southern edge of the navigational channel. It has only been in operation since February 2013 and is characterized by higher salinity values (approx. plus 2 psu) and lower turbidity than at HPA-Elbe 1. From figure 10 it can be seen that as at the other two stations salinity decreases during the 2013 river flood, but there is an increase in turbidity at the peak of the flood. This is not a spurious effect, since the helicopter transect 7 shows that the position of ETM at that time is at the site of MPM Otterndorf. In the course of the following two months, the reduced runoff leads to an intrusion of coastal water and an up-estuary movement of the ETM. At station LZ2 the storm surge on December 6 resulted in an intrusion of coastal water with an increase of salinity and, at this site, also an increase of turbidity.

Figure 10: Overview of the boundary conditions (runoff, wind speed) and measurements of salinity and turbidity at stations MPM Otterndorf. In the panels for wind speed, salinity, and turbidity tidal averages (solid black lines) and tidal extremes (grey shaded areas) are displayed. In the lowest panel near surface salinity (solid black line) and suspended particulate matter concentration (SPM, grey shaded curve) along the estuary from the 4 helicopter transects in 2013 are shown. The vertical line marked MPM indicates the position of the station. The dates of transects are also indicated as vertical lines in the time-series plots in the upper panels.

#### 4.4 Seasonal Variations

Due to the lack of data from the winter half-year no clear seasonal trends could by identified in the salinity and turbidity data at HPA-Elbe 1. Salinity is mostly influenced be episodic hydrological and meteorological events. Turbidity data in 2012 show a decrease during spring season, which is not observed in 2013 (figure 6). Even at LZ2 where data from the winter

season are available, no evident seasonal variations are obvious. Only water temperature and dissolved oxygen (depleted by higher biological degradation rates during the warmer season) show obvious seasonal variations.

#### 4.5 Spring-neap variations

- Spring-neap variations in the measurements at HPA-Elbe 1 are not particularly evident from figures 6 and 8. Distinct spring-275 neap variations are observed in the tidal water levels and the residual currents, the tide averaged current velocities. The timeseries of tidal range and residual currents at HPA-Elbe 1 during 2013 are displayed in figure 11. Spring tides in the Elbe Estuary occur with a lag of 2-3 days after full and new moon. The position of HPA-Elbe 1 is ebb dominant and residual currents are always positive with a range between 0.4 m s<sup>-1</sup> and 0.9 m s<sup>-1</sup>. Residual currents were stronger at springs than at neaps by a factor of 1.06 to 1.35. There is also some seasonality in tidal range and residual current at HPA-Elbe 1 with
- maximum values in summer and minimum values during winter.

#### Figure 11: Spring-neap variations of tidally averaged current velocity (grey shaded curve) and tidal range (solid black line) at HPA-Elbe 1. Vertical lines indicate dates of full (○) and new moon (●) (solid lines) and first and last quarter (dashed lines).

- Further spring-neap variations can be detected in the salinity measurements. The time-series of tide averaged salinity and 285 tidal range displayed in the upper panel of figure 12 do not always show a clear correlation, probably because spring-neap variations are superimposed by meteorological influences. A somewhat better correlation exists between the tidal salinity range (difference of maximum and minimum tidal salinity) and tidal range. The magnitude of the tidal salinity range is determined by the longitudinal estuarine salinity gradient at the station and the tidal excursion, which in turn is dependent on the current velocity. Since current velocities are highest during springs, there is a positive correlation between tidal salinity
- range and tidal range.

Figure 12: Spring-neap variations of salinity at HPA-Elbe 1. Upper panel: tidally averaged salinity (grey shaded curve) and tidal range (solid black line). Lower panel: Tidal salinity range (difference of maximum and minimum tidal salinity). Vertical lines indicate dates of full (0) and new moon (•) (solid lines) and first and last quarter (dashed lines).

- In contrast to previous measurements at other locations in the Elbe Estuary (Kappenberg et al., 1996), no clear correlation between turbidity (SPM) and tidal range is evident from figure 13. However, clear spring-neap variations appear in the difference between maximum flood values and maximum ebb values of turbidity, shown in the lower panel of figure 13. Lower values of this difference indicate a reduced tidal asymmetry in turbidity or a more homogeneous distribution of turbidity over the tide. These lower values coincide with springs when higher current velocities keep the particles in
- suspension over most of the tidal cycle.

Figure 13: Spring-neap variations of turbidity at HPA-Elbe 1. Upper panel: tidally averaged turbidity (grey shaded curve) and tidal range (solid black line). Lower panel: Difference of maximum turbidity value during flood and ebb. Vertical lines indicate dates of full (°) and new moon (•) (solid lines) and first and last quarter (dashed lines).

#### 305 4.6 Tidal variations

Tidal patterns in the measurements at HPA-Elbe 1 are already visible in figure 5. Under constant hydrological and meteorological conditions these patterns are repeated with little modification every tidal cycle until some forcing condition changes. This can be exemplarily be seen in figure 5 by the impact of the river flood on June 10, when the tidal patterns of salinity and turbidity change in the course of some tides.

- To define a characteristic set of tidal patterns for the measured variables for a certain episode an averaged tide was constructed. Again, as in section 4.1 a tide is defined by consecutive high water slacks, when the axial current velocity (that means the horizontal velocity component along the stream axis of the estuary) changes from flood to ebb current. Since the duration of the tide as well as relative durations of the ebb and flood-tide vary over a tidal cycle, the averaging process for different tides is not straightforward. To generate the averaged tide the following procedure was applied. First, the lengths of
- the averaged ebb and flood-tide were calculated as the average of all ebb and flood-tides in the ensemble. Interpolated values were than calculated for each variable at equidistant fractional steps (phases of the ebb/flood) separately for the ebb and flood part of every tide. These interpolated values were then averaged over the ensemble of tides. Finally, the phases of the ebb and flood-tide were multiplied by the lengths of the averaged ebb and flood-tides. Some results of these calculations are shown in figures 14 to 17 for a spring-neap episode in March 2012 under almost constant river runoff and weather
- conditions.

The variables water elevation, salinity, and turbidity are treated as functions of the current velocity and the corresponding tidal trajectory is displayed in the lower panel of the figures. 12 points at equal time intervals are indicated to demonstrate the rate of progress at different sections of the trajectory.

- In figure 14 the tidal patterns for (axial) current velocity and water elevation (above the instrument package at HPA-Elbe 1) 325 are displayed in the upper panel. There is a strong ebb dominance at this station expressed in a ratio of 2.3 between peak ebb and flood velocities and an ebb phase longer than the flood by a factor of 1.5. The ebb dominance is also visible in the lower panel where most of the trajectory is in the positive right ebb part. LW is near point 7 almost one hour before current reversal while HW occurs only 15 minutes before high water slack. There is also a characteristic slack asymmetry: current reversal happens much faster at LW than at HW.

# Figure 14: Tidal variation of axial current velocity and water elevation at HPA-Elbe 1. Upper panel: Averaged time-series of a spring-neap cycle (28 tides). Lower panel: Tidal trajectory of the averaged variables in current velocity/elevation space.

The tidal pattern of salinity at HPA-Elbe 1 is depicted in figure 15. Characteristic for this site is the occurrence of the salinity maximum 1.6 hours after high water slack during seaward-directed currents. This is probably the influence of saline water, 335 which was covering the wadden area to the North during the previous flood-tide. For over 4 hours around low water slack there is almost no variation in salinity. Until 1.8 hours after current reversal the salinity is not rising. Ebb dominance is here also expressed in the trajectory of salinity.

### Figure 15: Tidal variation of axial current velocity and salinity at HPA-Elbe 1. Upper panel: Averaged time-series of a spring-neap cycle (28 tides). Lower panel: Tidal trajectory of the averaged variables in current velocity/salinity space.

The tidal pattern of turbidity at HPA-Elbe 1 in figure 16 shows comparable peak values during ebb and flood-tide. Turbidity levels remain high at some 500 NTU while current velocities exceed a critical value of 0.5 m s<sup>-1</sup>, which is the case for most of the ebb and only a short period during flood-tide. While in the case of salinity the patterns can be explained exclusively by the advection of water masses with different salinities by the tidal currents, turbidity can also be produced and destroyed locally by resuspension and deposition of suspended material. Critical values of the current velocity exist, which generate critical turbulent shear stresses for resuspension and deposition of the fine-grained cohesive material engendering the turbidity. Critical current speeds for resuspension are encountered 1.5 hours after HW slack and because of the higher acceleration of the current already 40 minutes after LW slack. During the tidal cycle, turbidity never decrease below 170 NTU. This indicates that enough turbulence persists during the slack water periods to keep the sediment in suspension at the height of the sensor 1.5 m above the bottom. The trajectory of the turbidity in the lower panel of figure 16 indicates a lesser degree of ebb dominance than in the case of water elevation and salinity. It shows a typical 2 loop structure like a figure 8 rotate by 90 degrees. The crossing point of the trajectory is in the (positive) ebb section of the figure signalling ebb dominance at this site.

#### Figure 16: Tidal variation of axial current velocity and turbidity at HPA-Elbe 1. Upper panel: Averaged time-series of a springneap cycle (28 tides). Lower panel: Tidal trajectory of the averaged variables in current velocity/turbidity space.

As an example for the distinctiveness of the tidal patterns at different locations the current velocity and turbidity patterns at LZ2 only 4 km up-estuary of HPA-Elbe 1 are shown in figure 17. Here the pattern of current velocity is almost symmetrical and durations of ebb and flood-tide are equal. There is a characteristic 2 peak pattern of turbidity during flood-tide but the

- peak values during ebb and flood are almost the same. The asymmetry in the acceleration of the currents during HW and LW slack water is not as pronounced as at HPA-Elbe 1 and the resuspension of sediment commences 1 hour after HW slack and some 50 minutes after LW slack. Turbidity at HW slack is lower than at LW slack, which might be attributed to weaker turbulence at this site which gives the sediment more time to settle during the longer (compared to LW slack) HW slack
- water period. Also at this station, turbidity increases much faster after LW slack by rapid resuspension of sediment. However, when the local deposit is exhausted and carried up-estuary by the flood current, turbidity decreases again until after mid flood a second increase is observed, which might be attributed to suspended sediment advected from down-estuary reaches. During ebb-tide there is a steady rise of turbidity when first local deposited material is resuspended and later a never ceasing input of highly turbid water from the ETM located up-estuary of the station generates rising turbidity at LZ2. By the end of
- the ebb when current speeds fall below 0.5 m s<sup>-1</sup> deposition of sediment takes place, which depletes the water column of turbidity.

## Figure 17: Tidal variation of axial current velocity and turbidity at LZ2. Upper panel: Averaged time-series of a spring-neap cycle (28 tides). Lower panel: Tidal trajectory of the averaged variables in current velocity/turbidity space.

<sup>355</sup> 

#### 375 5. Discussion and Conclusion

Long-term measurements at fixed positions can be a crucial contribution to the understanding of estuarine processes. Although the site of the deployment is often a compromise between nautical and administrative on the one hand and scientific reasons on the other hand, the wide range of time-scales, which are involved in these processes, can only be investigated by such long-term measurements with a high temporal resolution. Their main disadvantage is that they are generally point measurements at a fixed place and in a certain depth. It is important that these point measurements are discussed and interpreted in the overall estuarine context and local peculiarities are not generalized. For example, the tidal current velocity as observed at HPA-Elbe 1 with its strong ebb dominance is not representative for region in the vicinity of the station, as can be seen from the comparison with data at LZ2. Whenever possible, the stations should be deployed and operated during the entire year to better investigate seasonal trends and the effects of storm and freshwater events, which most commonly occur during the winter half-year. Another serious shortcoming is the lacking vertical resolution. Although it can be speculated that given the ebb dominant near-bottom current velocity, the dominance should be even more expressed in the higher parts of the water column, important information on the vertical dynamics of salinity (mixing) and SPM is not available. So it is not possible from the time-series of turbidity in a single depth to distinguish between the effects of

turbidity shows recurrent patterns with two peaks at the phase of maximum ebb and flood currents. These can be attributed to the local resuspension of material deposited during the preceding slack water periods. The two peak patterns remain almost periodic for several tides before changing into a different pattern. This might be due to the relocation of local sediment deposits on the bottom. Spring-neap variations show up most clearly in the time-series of the hydrodynamic variables water elevation and current velocity. The greater tidal range at springs results in amplified currents. The resulting greater tidal excursion means that a longer section of the longitudinal salinity gradient passes the station and the range of salinity during a tidal cyle is wider. The same is true for the excursion of the ETM, but the range intratidal turbidity is also influenced by

horizontal advection with the tidal currents and vertical movement by settling and resuspension. The intratidal variation of

enhanced local resuspension by stronger currents.

The station measurements show a fair agreement with the data from the longitudinal helicopter transects in the vicinity of the stations. Although the transects give an excellent overview on the current status of the estuary there are too few of them and the temporal distribution is too irregular to allow for a quantitative analysis of seasonalities in the data. Compared to previous measurement in the ETM region of the central estuary the measurements from the stations in the outer estuary at the mouth of the Elbe display a more irregular behaviour. The impact of the meteorological forcing is much more important here than in the sheltered inner areas of the estuary.

A desirable extension of the investigations would be a comparison with results of numerical models. A first attempt was made using existing data from hydrodynamical model runs on a 50m by 50m grid for the year 2006. The modelled water levels in the grid cell containing HPA-Elbe 1 showed some agreement with tidal elevations recorded at the station, but the