# Peer review of "Seasonal, Spring-Neap and Tidal Variation of Hydrodynamics and Water Constituents in the Mouth of the Elbe Estuary, Germany"

_Ocean Science, 2016_

## Referee Comment (RC1) · Anonymous Referee #1 · 20 Jul 2016

Measurements of several parameters (e.g. water level, current velocity, turbidity, dissolved oxygen, salinity, wind) at a station named HPA-Elbe 1 in the mouth of the Elbe estuary are presented for the years 2012 and 2013. For comparison the measurements of two neighbouring stations (LZ2 and MPM Otterndorf) are used. To explain the overall estuarine context, data from surface samples taken by helicopter along a transect from the mouth of the Elbe to the weir Geesthacht are shown.

During the two years presented periods with low, high and extremely high (June 2013) river runoff are found. Periods with high westerly wind are found during winter.

For 9 days during the river flood in June 2013 all measurements at HPA-Elbe 1 are shown and discussed. In addition the measurements of salinity and turbidity respectively water temperature and oxygen saturation and the tidal average of the measured values are shown for 2012 and 2013. The comparison with the transect measurements of salinity and suspended particulate matter concentration respectively water temperature and dissolved oxygen saturation allows to compare the point measurements for several specific dates with the distribution along the estuary. The position of the mixing zone MZ and the estuarine turbidity maximum ETM are important for the understanding time series of the point measurements.

The measurements of salinity and turbidity and the tidal average of the measured values are shown for the neighbouring stations LZ2 (2012 and 2013) and MPM Otterndorf (2013), too.

All measured datasets are analysed in order to find seasonal, spring-neap and tidal variations of hydrodynamics and water constituents. The length of the time series did not allow for finding seasonal variations. Spring-neap variations were found in the measured water levels, current velocities and salinities. Tidal variations can be found in nearly all measurements. For 28 tides in March and April 2013 averaged time series of current velocity, water level, salinity and turbidity as well as tidal trajectories of the averaged variables are shown for station HPA-Elbe 1. For comparison averaged time series of current velocity and turbidity as well as tidal trajectories of the averaged variables are shown for station LZ2, too. The differences in the patterns observed are described but could not be explained without additional areal information.

The authors conclude that long-term measurements with a high temporal resolution at fixed positions are a crucial contribution to understanding of estuarine processes. The measurements should be representative for a larger region, which is difficult to guarantee in the area surrounding HPA-Elbe 1 with tidal flats and a branching river. As the authors could not explain all patterns found in the measurements they hope that results from numerical models of the Elbe estuary could help to close the gap between point measurements at different locations and the areal distributions of the physical quantities in the estuary.

General remarks

This paper gives an overview of measurements at three stations in the mouth of the Elbe during the years 2012 and 2013 describing the observed patterns in the measurements. The aim of the paper should be clarified more precisely. Please explain, why the station HPA-Elbe 1 was positioned in this area, what is the aim of the complex measurements at this position, why did you choose to investigate the variations on different time scales with the data of this station.

The quality of the measurements is not mentioned but should be discussed. The hydrological and meteorological situation should be described in addition to time series of river run off with e.g. time series of wind characteristic for the mouth of the Elbe (at Scharhörn) and time series of water level at the mouth of the Elbe estuary (situation of North Sea).

The measured quantities are presented in elaborated figures. Unfortunately due to the size of the figures it is difficult to detect all details mentioned in the text.

The severe river flood of June 2013 is mentioned in the abstract. All measured data are shown for this period, too. Explaining and analyzing the differences in the parameters measured before, during and after this extreme event should be a main emphasis of section 4. It would be interesting to see the development in time of the tidal trajectories during this period and compare these trajectories with the tidal trajectories of the averaged variables.

The authors hope for a better understanding of the measurements with the help of results from numerical modeling. Please discuss too, what kind of additional measurements could improve the understanding of the existing data, e.g. more stations, more trajectories along the Elbe during specific events, data on cross sections close to existing stations, etc.

I suggest a major revision of the paper before publication.

[Figure]

Specific points:

line 9: please do not use unexplained abbreviations.

line 45: specify the region where the large tidal range can be found.

line 49: . . . drains 132000 km2 until Neu Darchau.

line 53: wrong quote. Bergemann 1995 analyses the "Lage der oberen Brackwasser-grenze im Elbeästuar" (upper limit of the brackish water on the Elbe Estuary) and finds that for low discharges (< 400 m3/s) it moved upstream between 5 and 20 km between 1953 and 1994. Please correct your line of arguments.

line 83: "bei vollem Ebbstrom ca 1 h vor Tnw", is it correct to translated this as maximum ebb current?

line 90: section 2.2 Please explain (or refer to a document with a detailed description of the measuring campaign), how the measured parameters are converted into the parameters shown here, e.g. how is the water level or salinity measured und what is the expected error using this method.

line 111: figure 3 gives a rough overview of fresh water discharge and periods with operating stations for 2 years. In order to understand the hydrological situation the water level at the mouth to the North Sea (Bake A/Z or Cuxhaven) should be given too. "Higher water levels in the North Sea due to meteorological circumstances can generate higher salinity levels compared to the mean values of the lower Elbe (Boehlich and Strothmann, 2008)". Please add information about the water level of the North Sea.

line 112: the runoff in JFM2012 seems quite high. Why is 2012 described as a year with low runoff? Please give some statistical values supporting your argument.

line 115: a week of data is missing at HPA-Elbe 1?

line 124: Is the measurement at Scharhörn describing the meteorological / wind situation at HPA-Elbe 1? Please explain why you decided to use this place.

line 128 ff and figure 4: Do the years 2012 and 2013 show a characteristic distribution of wind speed and wind direction for this area of the Elbe? Why is there no high wind speed from 270 (west)? How many events with high wind speed were found in 2012 and 2013? Please explain why analysed data and not time series of wind are shown in this context.

line 138 figure 5: shows all parameters measured during the period of extreme discharge in June 2013. Please add the time series of Q at Neu Darchau and mention in the text the amount of time that the discharge signal needs to reach HPA-Elbe 1. Please give the reverence system for water level.

Is the wind measured at HPA-Elbe 1 or at Scharhörn? What is meant by "accidently" (line 155)? Is the wind strong enough to influence the measurements?

line 144: Flutstromkenterung - slack water time of flood current (DIN 4049-3 2.4.3.16) Please check translation of technical terms.

Line 168: here tide is defined as ebb current duration plus flood current duration?

Line 167 ff and figure 6: This figure gives a qualitative overview over 2 years of measurements. The details described in the text are hard to find due to the size of the figure. Please improve the figure. Why are transect 1 and 6 so different? What do transects look like in April 2012 and April 2013?

Line 175: Please give more detailed information, why the westerly winds and not any other process (e.g. higher water levels in the North Sea) cause this increase in salinity and up-stream shift of the turbidity.

Figure 7: Please describe the event flood June 2013 in more detail: Give the date for the points 1 to 4 (start increase of discharge in Neu Darchau, maximum discharge, . . .). How many days will it take for this signal to reach Geesthacht, Hamburg and HPA-Elbe 1?
Line 310 ff: Please give a reference for this method to determine the averaged tide. Please give some statistical evidence (standard deviation), that the 12 mean values are characteristic for this period in March / April 2013.

Line 360 and figure 17: which process produces the 2 maxima in turbidity during flood?

Technical remarks:

Line 33: please check: Dyer, 1977

---

## Referee Comment (RC2) · Anonymous Referee #2 · 27 Jul 2016

Summary

In this manuscript the authors present measurement data from three different long term deployments, ranging from spring 2012 to fall 2013 in the mouth of the Elbe estuary. These point measurements include CTD, velocity, turbidity and oxygen, as well as near by meteorological data. In addition nine along estuary helicopter transects are presented, with surface measurements of the same quantities as recorded at the fixed stations. Those transects provide snapshots of the along channel distribution.

Most of the analysis focuses on station HPA-Elbe1. The other two stations are primarily shown for comparison to provide an idea about the spatial variability.

The authors investigate the changes on different time scales, ranging from inter-tidal to seasonal variabilities. A special focus is put on an extraordinary strong discharge event in summer 2013.

General Comments

The authors present a very nice long term dataset, which taken by itself seems already worth publishing this manuscript. The language is mostly clear and the data are presented in a comprehensible manner. Unfortunately, the manuscript remains very descriptive based on a rather shallow data analysis. I understand that there is a trade off between showing such a large dataset as a whole and focusing on particular processes in detail. However, it would nice if there would be some further attempts to understand some of the underlying processes.

The authors claim in the first sentence of the discussion section, 'Long-term measurements at fixed positions can be a crucial contribution to the understanding of estuarine processes.' Although, I completely agree with that statement, I am not quite sure how the manuscript helps to improve our understanding of estuarine processes, since it only stays on a phenomenological level, barely touching any underlying processes and mechanisms. This is also reflected in the discussion section, which contains mainly general reflections on the usage of such long term data, without being able to provide any specific conclusion drawn from the actual observations. The discussion section should really focus more on the actual findings of paper rather than on general statements about the usability of long term point measurements.

The authors say that 'A desirable extension of the investigations would be a comparison with results of numerical models.' However it remains unclear what kind of questions the authors would like to address with a model that they can not investigate with this dataset.

I believe that the manuscript would benefit a lot from focusing on at least one of the observed phenomena in detail. For instance, you could focus a bit more on the high

discharge event, and how it effects the position of the ETM and the salinity gradient. While talking about the event you raise many questions that remain open. Why does it take so long to reestablish a 'fully developed ETM'? What does it mean that you have a relative 'quick recovery of turbidity' after the event, but it takes much longer for the ETM to come back to its normal state? What are the potential mechanisms that cause the hysteresis you observe in the salinity associated to the peak discharge event?

In general I think that the manuscript would be worth publishing after some major revisions.

Special Comments.

Section 4.1 and following. It is quite confusing to the reader to refer to the high discharge event as flood, especially in the context of inter tidal variability. Maybe you should try to avoid the term flood when referring to the river discharge.

142-143 'At the location of the station in the outer estuary, one would expect a more symmetrical tide and the asymmetry is probably due to influence of the branch (Medem Rinne) to the Northwest of HPA-Elbe 1.' Why would you expect a more symmetrical tide? Could you try to briefly discuss how the influence of the side branch could cause this asymmetry?

Almost all the figures show two different quantities in each panel (gray and black). It is a bit annoying to switch forth and back between figure and caption to be able to tell which quantity belongs to which color. It would be nice if you add a legend or a color coding to the labels off each panel.

257-259 The section title says 'Measurements at the MPM Otterndorf station', but here you only talk about LZ2.

296 Here you describe how the tidal asymmetry in turbidity changes with the Neap-Spring rhythm. However you do not try to explain what is the reason for the asymmetry in the first place. If I understand figure 13 correctly, positive values correspond to larger

peak flood than ebb values. This seems consistent with figure 5, however inconsistent with figure 16, where you find comparable peak values between ebb and flood. Am I missing something here? Furthermore, it seems surprising that you generally find larger peak values during flood than during ebb, given the fact that the site is clearly ebb-dominated with significantly larger peak ebb than flood current velocities. I feel this fact deserves some further discussion.

364 'Turbidity at HW slack is lower than at LW slack, which might be attributed to weaker turbulence at this site...' Why do you believe the turbulence should be weaker at this site? This is not obvious at all.

---

## Author Comment (AC1) · 13 Sep 2016

Dear referee,

I thank you for your comments on our manuscript. I generally agree with you that the purpose of the measurements at this site remains somewhat obscure. The main reason is that HPA initiated measurements on the northern wadden area (Neufelder Watt) some years before and in some way wanted to complete the general picture. The site is peculiar and I think no additional basic knowledge concerning estuarine processes can be gained from the time-series. A chance has been missed to install instruments (like ADCPs, sensor chains) which allow to study the vertical structure of the water column on a long-term basis. So the added value to the routine measurements at
the neighboring stations like LZ2 is rather low (also considering the gaps in the winter period).

I will follow your suggestions to concentrate on effects of the river flood in 2013 and discuss them in more detail, but this is impossible for me in the remaining time to the deadline. So this will actually lead to a new manuscript. The current manuscript will be withdrawn.

Thank you again and kind regards

Jens Kappenberg

—————————————————